# Effects of Inclusive Leadership on the Diversity Climate and Change-Oriented Organizational Citizenship Behavior

**DOI:** 10.3390/bs14060491

**Published:** 2024-06-11

**Authors:** JongHyun Lee, HyoungChul Shin

**Affiliations:** 1Graduate School of Tourism Management, Kyonggi University, Seoul 033746, Republic of Korea; jonghyun_l@naver.com; 2Department of Foodservice and Culinary Management, Kyonggi University, Seoul 033746, Republic of Korea

**Keywords:** inclusive leadership, diversity climate, organizational citizenship behavior, hotel industry

## Abstract

This study investigates the impact of inclusive leadership on diversity, climate, and change-oriented organizational citizenship behavior in hotel work. It also examines whether the diversity climate mediates the relationship between inclusive leadership and change-oriented organizational citizenship behavior. An online survey was conducted among hotel employees. It was found that inclusive leadership had a significant positive effect on the diversity climate. The diversity climate was found to have a significant positive effect on change-oriented organizational citizenship behavior, and inclusive leadership had a significant positive effect on change-oriented organizational citizenship behavior. Finally, inclusive leadership had a significant positive effect on change-oriented organizational citizenship behavior through a diversity climate. The results of this study have academic and practical implications for human resource management with respect to inclusive leadership in hotel workers’ work environment according to changes in hotels’ environmental factors for a new generation of employees flowing into the organizational mainstream.

## 1. Introduction

Hotel work is becoming increasingly complex and interdependent, and hotels play a more significant role in helping leaders acknowledge differences to foster a new culture in their employee relationships [1]. A hotel is where leaders and employees of an organization gather to discuss job performance. Effective performance can be challenging, as it includes visible differences such as gender, age, and race as well as invisible ones such as talent, education level, and cultural background in a working group [2,3]. In other words, if diversity is recognized, the organization and its employees will perform their duties effectively. At present, hotels are witnessing a new generation of employees entering their overall mainstream, and they are showing different attitudes and characteristics from those of previous generations, leading leaders to experience trial and error for the correct understanding and management of various values and behaviors [4,5,6].

Leadership leaders should have inclusive leadership that shows employees appreciation and encouragement for their contributions and their responses to an uncertain future through listening, effort, recognition, and response to individuals [7,8]. Inclusive leaders can change the working attitudes of employees, develop leadership and trust in the organization, and enable employees to act openly to communicate without barriers and form positive relationships [9]. Inclusive leaders seek to achieve desirable results by minimizing uncertainty, role stress, and anxiety among employees, strengthening commitment to individuals, and insisting on cooperation among team members [7]. Inclusive leaders’ supportive actions can maximize performance by encouraging employees to participate in decision-making, understand everyone from a position of equality, and encourage individuals to demonstrate their capabilities through work for the organization [10].

While there are some negative aspects to intellectual resource diversity, the positive aspects of diverse perspectives, information, and knowledge through the employment of personnel from different backgrounds have received greater emphasis [11]. Diversity in intellectual resources is related to the importance of organizational workforce diversity, and the greater the diversity of employees’ thoughts, skills, attitudes, and capabilities in the organization, the greater the company’s adaptability to a changing society and the greater the trigger for innovation [12]. Organizations must have access to a broader range of intellectual resources for achieving change-oriented innovation in a more efficient way [13,14,15]. Leaders who demonstrate inclusive leadership, a form of relational leadership, focus on the needs and difficulties of employees of an organization with diverse intellectual resources and act to help them [7,8,16]. Where both uniqueness and belonging are satisfied, employees can feel comfortable being their true selves [17,18]. Change-oriented non-work behavioral division [13,14]. contributes to an organization by demonstrating their potential by challenging new things [16,19].

Leaders should develop relational leadership and inclusive leadership to voluntarily perform challenging, change-oriented organizational citizenship behaviors that are difficult for employees to implement outside of the tasks that they are facing. This is because it is difficult for individuals to respond quickly to crises during rapid changes in the organization’s business environment [14,20]. From this perspective, inclusive leadership, or leaders’ relational leadership, is an essential factor that can improve organizational change and performance through the creation of a diversity climate, leading to change-oriented organizational citizenship behavior, an out-of-role behavior that is given to employees [14,21,22].

In spite of the importance of inclusive leadership, a climate of diversity, and change-oriented organizational citizenship behavior, studies that focus on the link between inclusive leadership, diversity climate, and change-oriented organizational citizenship behavior related to hotels are insufficient. In recent years, as the pace of change in society has accelerated, hotels are facing uncertainties and changes that have not been experienced before [23]. Various factors, such as technological advances, market fluctuations, and changes in consumer trends, change rapidly, and it is difficult for hotels to remain competitive if they fail to adapt to these changes [23]. Therefore, it may be necessary to identify the role of inclusive leadership, diversity climate, and change-oriented organizational citizenship that can induce the ability to quickly grasp and understand change, the ability to create and experiment with new ideas, and the ability to effectively communicate and implement change to members of the organization [23,24,25,26]. Despite the emphasis on the importance of inclusive leadership and diversity climate, few studies have applied and observed the relationship between inclusive leadership and diversity climate in the hotel industry. This study aims to empirically verify the effect of the leader’s inclusive leadership on the organization’s diversity climate and organizational citizenship behavior. In this relationship, we also examine the mediating effect of the diversity climate on inclusive leadership and organizational citizenship behavior. These empirical studies investigate the importance of inclusive leadership in forming a diversity climate and organizational citizenship behavior in hotels. Furthermore, based on the leadership’s inclusive leadership, we propose implications for maximizing the efficiency of human resource management in hotels.

## 2. Theoretical Background

### 2.1. Inclusive Leadership

Diversity management in organizations can be challenging, but inclusion in the workplace begins in relation to the perspective of viewing diversity in an organization as positive rather than negative [27,28]. Diversity management focuses on expanding opportunities for traditionally marginalized employees of organizations to work. It provides equitable opportunities for previously marginalized individuals to participate in work through the perspective of fairness [29]. The value of inclusion has been consistently discussed in this context, and it creates conflict resolution and harmony among individuals and groups in culture, society, and politics [30,31]. In other words, inclusion can be understood to extend beyond the perspective of discrimination-fairness to integration-learning [16,32,33].

Leadership research has shown dramatic developments, and transformational, ethical, empowering, genuine, and participatory leadership styles are also very diverse [34]. These leadership styles can help an organization [33]. In recent years, inclusive leadership that can create a diverse organizational inclusive climate has attracted attention. [16,33,35]. This leadership type is attracting attention among many scholars with respect to the relationship between leaders and employees in an organization. The influence of inclusive leadership based on social exchange theory has been shown to lead to improved perceptions, attitudes, and performance related to individuals’ work [7,10,30,31]. Scholars have clarified the concept of inclusive leadership, which can help maximize individuals’ potential through the enhancement of various employees’ uniqueness and sense of belonging, arguing that more research is needed [16,19].

Inclusive leadership maximizes the organization’s adaptability by improving the organization’s collective intelligence through an increase in the organization’s human and social capital. Inclusive leadership forms a process in which leadership demonstrates normatively appropriate behavior through interpersonal relationships or individual behavior, helping followers demonstrate this behavior in the processes of communication, motivation, and decision-making, Then, individuals exert influence on many employees for the achievement of common goals [36,37]. In inclusive leadership, supervisors create influence together rather than showing leadership to employees. This approach encourages followers of a leader to provide their value, inspiring a sense of belonging to an organization as employees [7].

Openness, availability, and accessibility are central to the composition of inclusive leadership [7]. Openness refers to the behavior of leaders who provide ideas or encourage their followers to participate in decision-making. Psychological safety can be enhanced through the formation of a work environment that is highly receptive to differences between employees [7,9]. Availability means that leaders can listen to their followers’ needs and questions as well as receive their feedback, and the degree of this feature can be checked by whether employees consider that they can discuss problems with leaders when they arise [7,9]. Finally, accessibility refers to the ability followers have to easily discuss necessary work issues with their leaders and the degree to which leaders are seen as easy to reach for discussing critical work issues [7,9].

### 2.2. Diversity Climate

Diversity is defined as individuals and groups of two or more, not at the individual level [11]. Identity in a group is an essential factor in diversity, and people tend to categorize themselves or others in relation to significant demographic characteristics, including gender, race, and ethnicity [38]. Traditionally, awareness of a climate of diversity refers to an individual’s perception of whether an organization is integrating various employees through their policies or strategies, such as in terms of minority employment preference policies and career development programs, and it can be measured as the subjective evaluation of whether the given organization values diversity [2,39]. Thus, in a traditional sense, the concept of diversity climate focuses on creating fairer workplaces by providing opportunities to minorities who were not recognized as mainstream employees and were omitted by organizations with respect to the aspects of discrimination and fairness [29].

Recent studies highlight several theoretical gaps and potential boundary conditions in the diversity climate and outcome relationship [2,40,41]. Existing studies have suggested that diversity climate affects performance through attitude perception [42], but many researchers have ignored this proposal and theorized direct relationships. Moreover, companies emphasize specific climate-related targets, such as services and safety, as strategic imperatives [40]. To maximize the potential of all of the organization’s employees and link them to performance, a limit has been identified to emphasize fairness or equity between mainstream and nonmainstream employees, which has been stressed through existing diversity climate studies. This has been expanded to reflect many subjective concepts, such as employees’ values, perspectives, thoughts, and attitudes [2,43]. It is necessary to include a process that synergistically promotes higher performance by effectively using human diversity in the organization. It is essential to understand the mechanisms of change in attitudes and behaviors of employees in relation to integration-learning, in which other employees’ views, backgrounds, knowledge, and ideas are actively exchanged [2]. In relation to synergy or integration-learning, the perception of diversity climate is the perception of employees of the degree to which the organization strives to listen to the opinions of various employees, recognize their values, and integrate multiple views to improve learning and performance in the organization, as well as fair treatment and personnel policies [2]. In addition to legitimacy in terms of fairness, equity, and discrimination, scholars indicate that recognition of differences, beginning with the concept of inclusion and recognition of diverse views, must be included [2]. Their diversity climate indicates the way that organizations accept different social statuses, such as those pertaining to gender and race [44].

A diversity climate must include consideration of minority groups, recognizing how valuable an organization’s employees are, promoting organizational diversity, and socially integrating employees belonging to marginalized groups [45]. Thus, the diversity climate has already produced positive value in its essence, such that the higher the degree, the more positive the organization appears [46]. A fair and inclusive climate is the central component of a diversity climate [21]. First, the development of a diversity climate must include various forms of fairness as perceived by employees when they work. This approach enables employees to become more identified with the organization and become more likely to commit to the organization while increasing productivity by performing their duties more extensively [47,48]. Second, the inclusion diversity climate can effectively predict satisfaction, immersion, and performance due to the recognition of how much the organization, through its management, is striving to achieve its mission, in which employees feel themselves part of a critical process in the organization, including their influence on its decision-making process, involvement in its essential work groups, and access to its information and resources [27,49,50,51,52].

### 2.3. Organizational Citizenship Behavior

Organizational citizenship behavior refers to actions that members of an organization voluntarily perform to develop the organization, apart from official work norms or rewards. This action shows commitment and responsibility to the organization beyond the minimum role performance of members of the organization. It is greatly influenced by individual efforts and the organizational environment [53]. Organizational citizenship behavior can be a practical aspect of behavior that can improve an organization’s effectiveness without incurring additional costs [54]. Organizational citizenship behavior is defined as discretionary behavior that promotes organizational efficiency, and has been classified into seven sub-elements: compliance, personal initiative, helping behavior, civic spirit, sportsman’s spirit, and loyalty to the organization [22]. That is, helping is the most frequent behavior and easiest to implement in organizational citizenship behavior. However, while behaviors including speaking and suggesting individual initiatives improve performance on the job, they are more likely to harm social relations within the organization because they may challenge customary workplace practices [22].

In recent decades, organized citizenship behavior has become one of the most popular topics in the public administration field [55,56,57]. However, some scholars have argued that organized citizenship behavior is insufficient to ensure an organization’s sustainable viability [14,53,58]. Organizations have suggested that change-oriented organized citizenship behaviors of members of organizations that tend to challenge the current state and bring about changes in organizational function are necessary [53,59]. It is explained that if an individual shows a high level of change-oriented organizational citizenship behavior, they are likely to be in a dangerous situation, mainly because it challenges the tradition and function of the organization in the organizational and political environment [53,59]. Employees with change-oriented organizational citizenship higher than general organizational citizenship behavior are expected to change organizational function about effectively working in the context of their job, unit of work, or organization [53]. Thus, challenging behaviors that are particularly difficult to implement in organizational citizenship behaviors are considered change-oriented organizational citizenship behaviors. 

On the basis of this perception, change-oriented organizational citizenship behavior is to be considered a voluntary and constructive effort made by employees to bring about new approaches to work, process changes, and policy changes to improve a work environment and its practices [14]. Because most existing variables related to change-oriented organizational citizenship behavior relate to organizational effectiveness, previous studies have generally studied the factors that influence change-oriented organizational citizenship behavior, while some theoretical reviews of compositional validity have also begun to be conducted [13,14]. Related studies show that leadership, leadership support, organizational support, and organizational atmosphere are widely adopted as antecedent variables for change-oriented organizational citizenship behavior [13,14,60]. Change-oriented organizational citizenship behavior is followed by change-oriented organizational citizenship behavior. The components of change-oriented organizational citizenship behavior are as follows: task modification [61], personal initiative [62], positive and active behavior [63], speech behavior [64], leading behavior [58], adaptation performance [65], creative performance [66], and innovative behavior [67].

### 2.4. Correlation among Variables

Research showing the mechanisms for the positive effects of the climate of atmospheric diversity within the organization in the relationship between leaders and employees is insufficient. Researchers who have previously studied diversity climate awareness have indicated the importance of organizations and leaders sending consistent messages to employees to reinforce the positive effects of diversity climate [68,69,70]. Leaders provide encouragement, support, and coaching to employees while also allowing them to participate in decision-making, creating a psychological climate, an environmental characteristic of the organization, to provide clues on the appropriateness and acceptability of specific actions for organizational salvation [19,71]. Where the inclusive leadership of leaders is successfully recognized among the organization’s employees, it can be adopted as a significant predictor of organizational performance through enhancing an organizational atmosphere of fairness and inclusion [7,8,16,17,18]. Ultimately, openness, usefulness, and accessibility, which embody inclusive leadership within the organization as perceived by employees, directly affect the organization’s climate of fairness and diversity [7,8,16,17,18,19,68,69,70,71]. This study established the following hypothesis, which implies that inclusive leadership affects the diversity climate.

**H1.** 
*Inclusive leadership has a significant positive effect on the diversity climate.*


There has been insufficient research describing the mechanisms for the positive effects of change-oriented organizational citizenship behavior in a climate of diversity in an organization and the relationship between employees. Previous studies indicate that the formation of a diverse atmosphere positively affects organizational citizenship behavior and organizational effectiveness by means of satisfaction, immersion, fairness, and friendly relationships, which are antecedent variables for organizational citizenship behavior. Diversity in the organization can be expected to present voluntary assistance and non-role actions for one’s fellow employees [72,73]. Employees believe that an organization’s ability and results are valuable if the price for their efforts is fairly distributed and embraced; if they perceive themselves to be helpful within the organization, they are all the more likely to take various actions to improve organizational performance [35,74]. If employees become fairly and inclusively aware of the atmosphere within the organization, the results of change-oriented civic organizational behavior can be enhanced and used as a significant predictor for organizational performance [13,14,16,19,75,76,77]. Ultimately, the climate of fair and inclusive diversity, which forms a climate of diversity in the organization as perceived by employees, directly affects change-oriented organizational citizenship behavior [13,14,16,19,35,72,73,74,75,76,77]. Synthesizing the above discussion, it is inferred that a diversity climate will affect change-oriented organizational citizenship behavior and establish the following hypothesis.

**H2.** 
*A diversity climate has a significant positive effect on change-oriented organizational citizenship behavior.*


Too little research has been conducted on the mechanisms for the positive effects of change-oriented organizational citizenship behavior on the relationship between inclusive leadership and organizational resources. However, such research has revealed the mechanisms for the positive effect of inclusive leadership and change-oriented organizational citizenship behavior in terms of the relationship between leaders and organizational employees [60]. Leaders’ inclusive leadership differs with regard to coaching and participatory leadership in its open nature, and such a leader’s behavior recognizes employees as a factor that drives innovation through the diversity of organizational employees and encourages proposals [9,78]. Inclusive leadership enhances the organizational adaptability of the organization’s employees by presenting ’their responsibilities and activities and promoting change and continuous innovation in the organization [79,80]. Leaders’ inclusive leadership can be understood as improving team innovation by providing workers with the resources, freedom, independence, and discretion necessary to perform their duties instead of commands and controls in terms of supporting employees [7,8,81,82]. Here, openness, usefulness, and accessibility, as well as inclusive leadership within an organization as perceived by employees, directly affect change-oriented organizational citizenship behavior [7,9,60,78,79,80,81,82]. This study synthesized the above discussion to infer that inclusive leadership can affect change-oriented organizational citizenship behavior and established the following hypotheses.

**H3.** 
*Inclusive leadership has a significant positive effect on change-oriented organizational citizenship behavior.*


**H4.** 
*Inclusive leadership has a significant positive effect on change-oriented organizational citizenship behavior through a diversity climate.*


This study aimed to investigate the effect of inclusive leadership on diversity climate and change-oriented organizational citizenship behavior (Figure 1).

## 3. Materials and Methods

### 3.1. Data Collection and Method

The participation sample in this study consisted of employees working at domestic hotels in 2023. Samples were collected using snowball sampling on 1 April 2023. The survey was completed through Google Online, and the link to the study was shared with ten collaborators working at the hotel through online messengers. Each hotel collaborator was provided with a prescribed product, and respondents collected data with voluntary consent. The first question in the online questionnaire was whether or not to participate in the survey, and it was conducted anonymously, so ethical approval was not required by Article 33 of the Statistical Act. The collected data were analyzed using SPSS 27.0 and AMOS 27.0. It was analyzed using a two-step approach, and confirmatory factors and reliability analysis were performed to verify the validity and reliability of the data. Structural equation modeling was performed to test the research hypothesis.

### 3.2. Measurement

The measurement items used in this study were drawn from organizational and human resource management studies. All items were translated and modified, and the participants provided their responses on a 5-point Likert-type scale (from strongly disagree to strongly agree).

Inclusive leadership was measured in terms of openness (four items), availability (four items), and accessibility (four items) [7,83]. The parameter of diversity climate was measured in terms of fair diversity climate (five items) and inclusive diversity climate (four items) [21]. Finally, the dependent variable, change-oriented organizational citizenship behavior, was measured using four questions [14,58,67]. Detailed measurement questions for variables are in Appendix A.

## 4. Results

### 4.1. Demographics of the Participants

Table 1 presents the demographic characteristics of the sample. Men were 52.0% of the sample, and women were 48.0%. The largest group by age was those in their 20s, at 70.2%, and the most common position was that of an associate administrator, at 29.8%. The most common working period was less than 3 years, at 67.4%.

### 4.2. Reliability and Validity

#### 4.2.1. Confirmatory Factor Analysis

The results of the confirmatory factor analysis conducted to assess the feasibility of the research structure are presented in Table 2. Inclusive leadership and diversity climate are composed of secondary factors, and the sub-variables for each concept were measured as primary factors by averaging their mean points using a multiple-dimension bundle approach. The results indicate the fitness of a three-factor model (i.e., inclusive leadership, diversity climate, and change-oriented organizational citizenship behavior) for the data, based on goodness-of-fit statistics. A good model fit was assessed, as follows χ^2^ = 46.507 (df = 24, *p* = 0.000), CMIN/DF = 1.938, RMR = 0.010, GFI = 0.973, AGFI = 0.949, NFI = 0.990, IFI = 0.990, TLI = 0.985, CFI = 0.990, RMSEA = 0. In addition, the standardized factor load was 0.5 or higher, and concept reliability was statistically significant (0.7 or higher) or better [84]. Cronbach’s alpha was used to measure the level of internal consistency of each structure, and the values of 0.870 and 0.911 met the threshold [85].

#### 4.2.2. Discriminant Validity

Table 3 presents the discriminant validity. Applying the relationship between diversity climate with the highest correlation between variables and change-oriented organizational citizenship behavior, the correlation coefficient between diversity climate and change-oriented organizational citizenship behavior was 0.731; (0.731)^2^ = 0.538. Therefore, the AVE for the diversity climate was 0.907, and the AVE for change-oriented organizational citizenship behavior was 0.780. Discriminant validity was supported because the AVE value of the two variables was larger than the square of the correlation coefficient, and the inclusive leadership AVE 0.895, which is greater than 0.538 [86].

### 4.3. Hypothesis Testing

The results support all hypotheses (H1: β = 0.778, *p* < 0.001; H2: β = 0.644, *p* < 0.001; H3; β = 0.242, *p* < 0.01). Table 4 presents the results of SEM computed using AMOS 27.0 to test the hypothesis. The model goodness-of-fit was χ^2^ = 46.507 (df = 24, *p =* 0.004), CMIN/DF = 1.938, RMR = 0.010, GFI = 0.973, AGFI = 0.949, NFI = 0.980, IFI = 0.990, TLI = 0.985, CFI = 0.990, and RMSEA = 0.051 [87]. Hypothesis 1 was supported (β = 0.730, *p* < 0.001). Hypothesis 2 was supported (β = 0.719, *p* < 0.001). Hypothesis 3 was supported (β = 0.143, *p* < 0.05). Hypothesis 4 was verified by applying 500 bootstrapping samples. For Hypothesis 4 was adopted (β = 0.525, *p* < 0.01).

## 5. Discussion

As work becomes more complex and interdependent and a new generation of employees is flowing into organizations’ overall mainstream, leadership is needed more than ever to strengthen the inclusive leadership that is perceived by the organization due to changes in the environmental factors that surround the hotel [88]. The systematic response of a hotel requires voluntary and dedicated participation and efforts from the hotel’s standpoint to increase the positive and inclusive leadership of the organization’s employees, which is expected to be affected by hotels’ leadership operations [88].

This study was conducted to provide data to describe the development of hotel leaders’ leadership through identifying the effect of positive, inclusive leadership as perceived by employees on diversity climate and change-oriented organizational citizenship behavior, the effect of diversity climate on change-oriented organizational citizenship behavior, and the mediating effects of diversity climate between inclusive leadership and change-oriented organizational citizenship behavior [24,25,26]. Hotels are an international environment where customers and organizational members from various cultural backgrounds meet, and the importance of inclusive leadership, diversity climate, and change-oriented organizational citizenship behavior is being emphasized more to maintain competitiveness in this environment [24,25,26]. The results of the empirical analyses are as follows: In the investigation of Hypothesis 1, it was found that inclusive leadership perceived by hotel employees directly affects the diversity climate. This finding supports the results of previous studies that inclusive leadership affects the diversity climate [7,8,16,17,18,19,24,68,69,70,71]. It is necessary to create a form that can reduce negative organizational culture by emphasizing a positive organizational atmosphere and recognizing the need for others’ inclusive leadership at the organizational level. It is possible to resolve leaders’ openness, usefulness, and accessibility by empirically verifying that they provide encouragement, support, and coaching to the employees of the organization and allowing them to participate in decision-making [24].

The investigation of Hypothesis 2 confirmed that the diversity climate perceived by employees of a hotel company had a direct effect on change-oriented organizational citizenship behavior, and the results of several previous studies were supported [13,14,16,19,25,35,72,73,74,75,76,77]. This may be the primary factor influencing smooth communication between the organization and its employees, namely, the basis for a hotel’s organizational performance. The climate of fair diversity and inclusive diversity of the organizational atmosphere perceived in the process of performing the entire capabilities of the leaders and employees of the organization is likely to contribute to the development of hotels regarding changes in the times by forming punitive and active behaviors that positively shape hotels’ organizational cultures. Change-oriented organizational citizenship behavior taking place for leaders in an organization and its employees can resolve the management method without development in response to a smooth organizational environment operation method [25].

The investigation of Hypothesis 3 showed that inclusive leadership, as perceived by the employees of a hotel company, had a direct effect on change-oriented organizational citizenship behavior, and the results of several previous studies were supported [7,9,26,60,78,79,80,81,82]. This is interpreted to mean that it is possible for employees to smoothly share an organizational atmosphere that is based on individual intellectual ability in hotel companies that have a relatively high human dependence to secure strong competitiveness. In hotels, trust and commitment are formed within the organization. Openness, usefulness, and accessibility, which form the inclusive leadership of leaders in the organization, require management that can reduce hostile leaders’ awareness of inclusive leadership to allow them to quickly move forward with rapidly developing trends of creative, innovative, and change-oriented organizational citizenship behaviors without concealing the unique intellectual ability between hotels and their employees [26].

The investigation of Hypothesis 4 confirmed that the diversity climate had a mediating effect on change-oriented organizational citizenship behaviors, and the results of several previous studies were supported [7,8,9,13,14,16,17,18,19,24,25,26,35,60,68,69,70,71,72,73,74,75,76,77,78,79,80,81,82]. It was found that inclusive leadership’s effect on diversity climate and diversity climate on change-oriented organizational citizenship behavior was more significant than its effect on change-oriented organizational citizenship behavior. Therefore, diversity climate is an important parameter. From this, we can conclude that the inclusive leadership of leaders who will be needed for hotels to evolve into a creative and innovative form may lead to a positive diversity climate while negatively affecting the organization. An inclusive leader of a hotel can create an atmosphere of respecting and embracing members of various organizations with leadership styles that encourage and appreciate the participation of members of various cultural backgrounds, values, and ideas and support all members of the organization to contribute to the achievement of organizational goals [24,25]. Furthermore, it can positively recognize organizational changes and induce the actions of actively participating members [24,26]. Therefore, it is necessary to manage the organizational atmosphere that can lead to a high tide in a happy and evolutionary atmosphere between hotels and their employees [24,25,26].

### 5.1. Theoretical Implications

We present the academic implications of this study. The inclusive leadership of leaders perceived by hotel employees was made up of openness, usefulness, and accessibility, and change-oriented organizational citizenship behavior was finally understood to be a single dimension including diversity climate factors, such as a fair diversity climate and an inclusive diversity climate. This study is the first to have investigated the management and development of the workforce of hotel leaders and employees who are highly dependent on others, relative to other industries, incorporating recent changes in environmental factors of the organizational atmosphere that surround intergenerational heterogeneity in hotels.

While previous studies focused on the simple correlation between inclusive leadership, diversity climate, and change-oriented organizational citizenship behavior variables, this study provided an in-depth understanding of the relationship, revealing that inclusive leadership creates a diversity climate and that diversity climate plays a mediating role in promoting change-oriented organizational citizenship behavior. Furthermore, by presenting a new perspective on organizational management research, research on the impact of the organizational environment on organizational citizenship behavior and the individual characteristics of organizational members was reinforced. Few scholars have so far studied the factors explaining the relationship between hotel-inclusive leadership, diversity climate, and change-oriented organizational citizenship behavior. There has been a lack of research on the impact of inclusive leadership of leaders and hotel employees with respect to diversity climate and change-oriented organizational citizenship behavior, the effect of diversity climate on change-oriented organizational citizenship behavior, and whether diversity climate mediates the effects between inclusive leadership and change-oriented organizational citizenship behavior. The importance of each aspect of inclusive leadership, diversity climate, and change-oriented organizational citizenship behavior was considered equally in determining the job performance capabilities of hotel leaders and employees. 

### 5.2. Practical Implications

We present the practical implications of this study as follows. In the deepening organizational culture of hotel companies, where conflicts between generations within organizations are increasing in a rapidly changing environment, it is necessary to reregulate the behavior and roles of leaders in response to changes in various environmental offices that surround hotel companies and to alleviate the organizational atmosphere within organizations so that it can be expressed as actions beyond change-oriented work based on cooperation and trust between hotels, leaders, and employees. This study suggests that the inclusive leadership of hotel leaders can increase organizational performance by leading to a diversity climate within the organization and change-oriented organizational citizenship behavior for the employees of the organization.

Hotels can increase change-oriented organizational citizenship behavior by creating a diversity climate that emphasizes inclusive leadership and fosters a positive organizational atmosphere about leaders’ leadership [7,14]. It is essential to create an organizational culture in which employees of the hotel can freely express themselves in the decision-making process [18]. Hotel managers and leaders should seek to create a happy organizational atmosphere in relation to inclusive leadership and increase change-oriented organizational citizenship behavior [8,75,80].

Hotels are more dependent on human resources than organizations in other industries. The intellectual ability and job competence of employees of a hotel, which are intangible assets, greatly affect customer satisfaction and organizational development, so respectful communication and organizational culture are needed [17]. The openness, usefulness, and accessibility of hotel leaders’ inclusive behavior should be accompanied by management’s creation of an organizational atmosphere of a fair diversity climate and an inclusive positive diversity climate, as well as ensuring that change-oriented organizational citizenship behavior can continue [8,13,16]. Through leaders’ leadership, which can lead to continuous trust between them and employees, the impact on leaders and employees can be understood [7,16]. However, excessive emphasis and intervention by hotel managers on inclusive leadership can negatively affect leaders [81]. Managers cannot obtain good results in terms of organizational performance if they force leaders to improve their intellectual and job competencies [82]. In addition, the excessive involvement of hotel managers negatively affects leaders’, pride and identity in creating a good organizational culture [19]. Therefore, hotels should invest in developing inclusive leadership to create a climate of diversity and promote change-oriented organizational citizenship behavior based on a positive climate of diversity. Through leadership development programs for leaders, inclusive leadership capabilities should be strengthened, the cultural backgrounds of various organizational members should be welcomed, and they should be helped them adapt to the organization. In addition, developing a strategy that can provide change education programs and establish a compensation system for organizational members who actively participate in change is necessary. It is necessary to prepare a plan for appropriate harmony by means of leadership management centered on smooth communication and organizational performance among managers, leaders, and employees [19,77,80].

## 6. Conclusions

In a hotel, human resources have significant value because hotel employees directly provide services, leading to smooth management. For this, inclusive leadership, which is the relationship-oriented leadership of hotel leaders, is required. Some hotels have encountered problems in terms of operation and workforce management for profit only, taking into account errors in communication between leaders and employees. This study investigated the influence of inclusive leadership on the part of hotel leaders on organizational atmosphere, diversity climate, change-oriented organizational citizenship behavior of hotel employees, and the mediating effect of diversity climate between inclusive leadership and change-oriented organizational citizenship behavior.

It was found that all variables of inclusive leadership, diversity climate, and change-oriented organizational citizenship behavior had positive effects, directly or indirectly, and they created an atmosphere that recognized and respected the diversity of employees of the organization and enabled an open and cooperative communication environment. Leaders promote creativity and innovation, and they improve immersion and a sense of belonging to the organization by collecting and using employee opinions. A positive perception of change among employees is thereby formed, forming a solid motivation for participation and cooperation in change. It is expected that employees will experience less uncertainty regarding the change process and further increase the likelihood of success with organizational change. This study is expected to make a contribution to the development of leadership and organizational culture theory by examining the influence of inclusive leadership on the diversity climate, which is an organizational atmosphere; the impact of diversity climate on change-oriented organizational citizenship behavior; and the mediating effect of diversity climate between inclusive leadership and change-oriented organizational citizenship behavior.

Based on the results of this study, the following research can be proposed. Research on how other organizational variables such as organizational culture, organizational structure, and leadership style affect the results of this study can be conducted, and it is believed that longitudinal studies can identify changes in inclusive leadership, diversity climate, and change-oriented organizational citizenship behavior variables over time. The limitations of this study are as follows: First, there may be differences in the impact relationship across hotels, depending on the given hotel’s management type, inclusive leadership, diversity climate, and change-oriented organizational citizenship behavior as perceived by leaders and employees of the organization. This includes leaders’ kind of inclusive leadership in the organization, the diversity climate, and the change-oriented organizational citizenship behavior of the employees of the organization. Therefore, it is necessary to determine how the hotel’s inclusive leadership, diversity climate, and change-oriented organizational citizenship behavior affect outcome variables and how these differ by type. Second, the research was conducted only on hotel leaders and employees. However, because each hotel’s organizational culture is different, with conservative or progressive values, inclusive leadership, a diversity climate, and change-oriented organizational citizenship behaviors recognized by leaders and employees of the organization may differ. Therefore, it is necessary to measure inclusive leadership, diversity climate, and change-oriented organizational citizenship behavior in a way that can be understood together in the rapidly changing trends of the times for leaders and organizational employees by dividing the hotel’s management form and organizational culture. Third, the snowball sampling method was valid when the study target group was not directly known, but efforts are needed to solve this problem due to sample representativeness problems. Therefore, collecting as much information as possible about the study target group and sampling it using various methods is necessary. Fourth, we failed to identify effective performance according to age, gender, race, and position differences. Therefore, it will be mandatory to consider modulating variables when evaluating the inclusive leadership study of hotel employees. Lastly, we used three aspects of inclusive leadership constructed in previous studies. Research on inclusive leadership is in its early stages, and based on the results of this study, we expect theoretical expansion in future studies.

## Figures and Tables

**Figure 1 behavsci-14-00491-f001:**
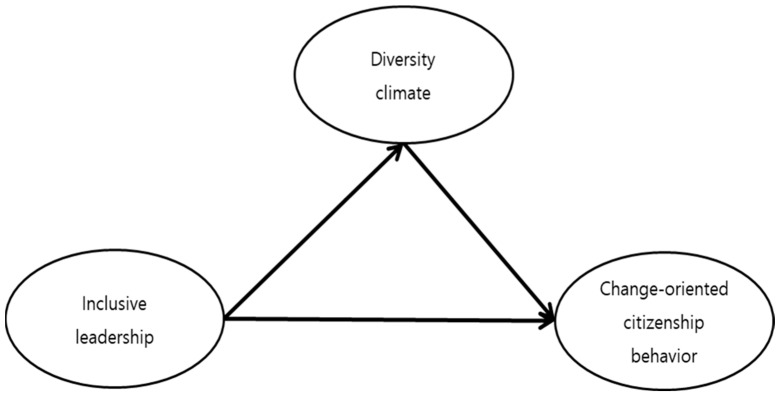
Study model.

**Table 1 behavsci-14-00491-t001:** Demographic characteristics of the participants.

Demographic Factors	Category	Number of Participants	Percentage (%)
Gender	Male	185	52.0
	Female	171	48.0
Age	20s	250	70.2
	30s	73	20.5
	40s	30	8.4
	50s and older	3	0.8
Education	High school diploma or less	64	18.0
	Associate degree	106	29.8
	Bachelor’s degree (4-year university)	104	29.2
	Graduate degree or higher	82	23.0
Working period	Less than 3 years	240	67.4
	Between 3 and 5 years	71	19.9
	Between 5 and 10 years	24	6.7
	More than 10 years	21	5.9
Total		356	100

**Table 2 behavsci-14-00491-t002:** Confirmatory factor analysis.

Factor and Variable		Standardized Loading	S.E	C.R	AVE	Composite Construction Reliability (CCR)	Cronbach’s α
Inclusive leadership	Openness	0.907	-	-	0.895	0.962	0.911
	Availability	0.856	0.043	22.356 ***			
	Accessibility	0.878	0.041	23.403 ***			
Diversity climate	Fair diversity climate	0.897	-	-	0.907	0.951	0.877
	Inclusive diversity climate	0.871	0.046	20.778 ***			
Change-oriented Organizational behavior	COOI1	0.761	-	-	0.780	0.934	0.870
	COOI2	0.809	0.068	15.519 ***			
	COOI3	0.851	0.069	16.366 ***			
	COOI4	0.757	0.072	14.406 ***			

χ^2^ = 46.507 (df = 24, *p* = 0.000), CMIN/DF = 1.938, RMR = 0.010, GFI = 0.973, AGFI = 0.949, NFI = 0.980, IFI = 0.990, TLI = 0.985, CFI = 0.990, RMSEA = 0.051. *** *p* < 0.001.

**Table 3 behavsci-14-00491-t003:** Discriminant validity of the variables.

Factor	Inclusive Leadership	Diversity Climate	Change-Oriented Organizational Behavior
Inclusive leadership	0.895 ^(1)^	0.423 ^(3)^	0.367
Diversity climate	0.651 **^(2)^	0.907	0.534
Change-oriented organizational behavior	0.606 **	0.731 **	0.780

Note: ** *p* < 0.01. (1) Diagonal represents the average variance extracted; (2) area below the diagonal represents the correlation coefficient for the constructs (r); (3) area above the diagonal represents the square of the correlation coefficient (r^2^).

**Table 4 behavsci-14-00491-t004:** Results of structural equation model analysis.

		Estimate	*t*-Value	*p*-Value	Indirect Effect		Decision
	Process (Hypothesis)				Estimate	*p*	
H1	Inclusive leadership -> diversity climate	0.730	14.505 ***	0.000			Accepted
H2	Diversity climate -> change-oriented organizational citizenship	0.719	9.173 ***	0.000			Accepted
H3	Inclusive leadership -> change-oriented organizational citizenship	0.143	2.123 *	0.034			Accepted
H4	Inclusive leadership -> change-oriented organizational citizenship (mediating effects of diversity climate)				0.525 **	0.004	Accepted

* *p* < 0.05; ** *p* < 0.01; *** *p* < 0.001.

## Data Availability

The data presented in this study are available on request to the corresponding author.

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
