# Peer review of "Effects of Inclusive Leadership on the Diversity Climate and Change-Oriented Organizational Citizenship Behavior"

_behavsci, 2024, doi:10.3390/bs14060491_

Round 1

Reviewer 1 Report

Comments and Suggestions for Authors

Strengths

- Originality or innovativeness of the manuscript.

- Clear message and clarity of argumentation.

- Manuscript structure.

- Well written and logical.

Weaknesses:

- The methodology approaches (provide clear and better explanations for the

methodology strategy).

- Conclusions (the conclusions are very concise and brief, lacking fundamental analytical elements).

- References (needs more current references)

The manuscript adequately adheres to the mission and scope of the journal. The topic is relevant and current and relevant to scientific research. Clearly define the overall objectives. It is well written and logical. Presents relevant arguments to justify the research as a whole and the issue under investigation. The paper is explicit about its implications for research, practice and/or society. Although the study was conducted in a coherent way, it presents problems that compromise its contribution to the scientific community.

 I summarize these problems:

(1) Review the writing (there are very long paragraphs throughout the article, namely in the introduction, literature review and discussion).

(2) The introduction does not include the necessary elements. It should be structured as follows: contextualization and relevance of the theme, objectives (general and specific), methodology, study contributions, article structure.

(3) The paper does not undertake a critical approach in reviewing the literature. Absence of theoretical foundation and the most current references of the main topics (the topic “2.2. Diversity Climate” has no references from the last 3 or 5 years; the topic “2.3. has no references from the last 3 or 5 years!). The literature review is not critical, reflective, and does not effectively show the state-of-the-art research in this area.

(4) The methodology approaches must be improved. Provide clear and better explanations for the research design. I suggest I suggest presenting the following topics: methodological strategy, information gathering techniques, population and sample, procedure, data collection techniques.

(5) The discussion of the results must be more robust should be more reflective, incorporating more authors references (more current) to support the analyzes/results achieved.

(6) The conclusions are incipient. The conclusions should be better discussed, based on contributions from some authors. Conclusions should also involve generalization of the comments of the authors regarding the issue, highlights, and research gaps. It should also present the next research steps.

Author Response

Dear reviewer,

We would like to express our sincere gratitude for your invaluable guidance on our manuscript. Your comments are crucial to us and have significantly contributed to enhancing the quality of our work. Following your feedback, we have made further revisions to the initial manuscript, with all changes highlighted in red. While the revised manuscript may still have some imperfections, we kindly ask for your ongoing suggestions to further improve it. Once again, we thank you for your hard work and patient guidance. 

Reviewer 2 Report

Comments and Suggestions for Authors

Literature review and discussion sections need significant work to appropriately define constructs and clarify meaning and purpose.

Comments on the Quality of English Language

English language is good, but there are errors throughout that need to be addressed. Some have been identified in the attached word document.

Author Response

(The authors gave the same response as above.)

Reviewer 3 Report

Comments and Suggestions for Authors

The theoretical framework, previous studies on the topic, and the development of the hypotheses are clearly presented. The empirical findings are described and discussed in detail. However, there is also potential for improvement:

Relatively few post-2020 publications were referenced. What exactly the research gap is that the study aims to close should be made clearer in the introduction. It should also be mentioned in the introduction that the adaptability of the organization is becoming more important because the environment is changing more strongly or more quickly. This is only explained at the end of the article. The same applies to some special features of the hotel industry. It would also be desirable to explicitly explain at some point how inclusive leadership differs from other leadership styles and what the relationship is between inclusive leadership and diversity management.

In chapter 2.3, it is appropriate to take up the definition of ‘organisational citizenship behaviour’ first.

In the ‘Materials and Methods’ chapter, the measurement items used should be listed in detail, e.g. in a footnote.

In the ‘Results’ chapter, table 1 refers to the educational degree, among other things, but the text refers to the professional position. This should be harmonised. The low β for H3 and H4 should also be discussed in table 4. The values in table 4 for H3 and H4 are identical. This should be checked.

Chapter 5 states ‘that the inclusive leadership of leaders who will be needed for hotels to evolve into a creative and innovative form may lead to a positive diversity climate while negatively affecting the organization’. A justification for the negative effect on the organization would be desirable.

Chapter 5.1 states: ‘this study explored the level of behavioral effects outside of the organization’. In my opinion, this external aspect of the organization has not emerged from the previous explanations.

In the ‘Limitations’ chapter, it would also be useful to take a closer look at the problems of snowball sampling, which was chosen for the study.

Author Response

(The authors gave the same response as above.)

Round 2

Reviewer 1 Report

Comments and Suggestions for Authors

Dear Editors,

Dear Authors,

Regarding the manuscript ID behavsci-3007239, entitled "Effects of Inclusive Leadership on the Diversity Climate and Change-Oriented Organizational Citizenship Behavior", I inform you that the changes are in line with my expectations.

I confirm that this new version is in line with my requirements. But, if possible, I suggest authors include the type of study at the beginning of point 3. Materials & Methods. This research configures a quantitative and correlational methodology (Field 2009) and is a cross-sectional research design (Ahiauzu & Asawo, 2016). Explain this…

Best regards,